# Targeting SLC7A11-mediated cysteine metabolism for the treatment of trastuzumab-resistant HER2-positive breast cancer

Yijia Hua[1,2†], Ningjun Duan[1,3*†], Chunxiao Sun[1], Fan Yang[1], Min Tian[1], Yanting Sun[1], Shuhan Zhao[4], Jue Gong[1], Qian Liu[1], Xiang Huang[1], Yan Liang[1], Ziyi Fu[1], Wei Li[1], Yongmei Yin[1,2,5*]

[1]Department of Oncology, The First Affiliated Hospital of Nanjing Medical University, Nanjing, China; [2]Gusu School, Suzhou Municipal Hospital, The Affiliated Suzhou Hospital of Nanjing Medical University, Suzhou, China; [3]Department of General, Visceral and Pediatric Surgery, University Medical Center Göttingen, Göttingen, Germany; [4]Department of General Surgery, The First Affiliated Hospital of Nanjing Medical University, Nanjing, China; [5]Jiangsu Key Lab of Cancer Biomarkers, Prevention and Treatment, Collaborative Innovation Center for Personalized Cancer Medicine, Nanjing Medical University, Nanjing, China

*For correspondence:
ningjun_official@outlook.com
(ND);
ymyin@njmu.edu.cn (YY)

†These authors contributed equally to this work

Competing interest: The authors declare that no competing interests exist.

## eLife Assessment

This study provides **compelling** evidence that SLC7A11 may serve as a potential therapeutic target for trastuzumab-resistant HER2-positive breast cancer. While the findings are well-supported by robust data, the study could have been further strengthened by incorporating additional cell line experiments and providing more detailed clarification on patient sample selection. Nevertheless, this **valuable** work represents a significant contribution and will be of considerable interest to researchers in the field of breast cancer.

**Abstract** Trastuzumab resistance remains a challenge for HER2-positive breast cancer treatment. Targeting metabolic reprogramming would provide novel insights for therapeutic strategies. Here, we integrated metabolomics, transcriptomics, and epigenomics data of trastuzumab-sensitive and primary-resistant HER2-positive breast cancer to identify metabolic alterations. Aberrant cysteine metabolism was discovered in trastuzumab primary-resistant breast cancer at both circulating and intracellular levels. The inhibition of SLC7A11 and cysteine starvation could synergize with trastuzumab to induce ferroptosis. Mechanistically, increased H3K4me3 and decreased DNA methylation enhanced SLC7A11 transcription and cystine uptake in trastuzumab-resistant breast cancer. The regulation of epigenetic modifications modulated cysteine metabolism and ferroptosis sensitivity. These results revealed an innovative approach for overcoming trastuzumab resistance by targeting specific amino acid metabolism.

## Introduction

Accounting for 15–20% of all breast cancer cases, human epidermal growth factor receptor 2 (HER2) positive breast cancer has posed great threats to the health of women worldwide (*Giaquinto et al.,*

*2022*; *Harbeck et al., 2019*). The overexpression of HER2 results in aberrant breast cancer cell proliferation, invasion, and progression (*Swain et al., 2023*). As a monoclonal antibody targeting HER2, trastuzumab has been recognized as the foundational medication in HER2-positive breast cancer treatment and provided remarkable clinical benefits for numerous patients (*Marra et al., 2024*). Around 50% HER2-positive breast cancer patients would experience trastuzumab resistance and disease progression during or after treatment settings (*Cossetti et al., 2015*). Current theories maintained that trastuzumab resistance mainly resulted from HER2 mutation (*Xu et al., 2017*), antigenic epitope variation (*Arribas et al., 2011*), compensatory pathway activation (*Baselga et al., 2014*) and tumor heterogeneity (*Marchiò et al., 2021*). However, there still lacks a comprehensive theory to thoroughly explain the mechanisms behind trastuzumab resistance, and exploring effective targets to overcome resistance remains a major topic in the field of breast cancer research (*Swain et al., 2023*).

Metabolic reprogramming has been acknowledged as one crucial cancer hallmark (*Hanahan, 2022*). Traced back to the first study carried out by Otto Warburg, decades of studies have indicated that glycolysis, tricarboxylic acid (TCA) cycle, lipid metabolism, amino acid metabolism, as well as other processes collectively modulate the development and function of cancer cells and tumor microenvironment (TME) components (*Martínez-Reyes and Chandel, 2021*; *Vander Heiden and DeBerardinis, 2017*; *Kao et al., 2022*). Several studies have revealed that different breast cancer subtypes featured diverse metabolic patterns (*Gong et al., 2021*; *Xiao et al., 2022*). Targeting specific enzymes or modulating nutrient uptake might normalize the TME and enhance breast cancer treatment efficacy (*Miller et al., 2023*; *Xiao et al., 2023*). Yet previous studies often focused on the overall metabolic subtypes of breast cancer, with little attention given to abnormal metabolic changes and underlying genomic alterations induced by specific drug resistance. This might lead to a poor understanding of the tumor heterogeneity in drug-resistant breast cancer cells.

Epigenetic modifications play crucial roles in all stages of cancer development and contribute to cancer cell survival selection (*Timp and Feinberg, 2013*). Alterations in chromatin structure, DNA methylation, and histone modifications provide a tumor growth advantage and increase phenotypic variability (*Feinberg et al., 2016*). During the formation of drug resistance, epigenetic dysregulation modulates key genes transcription associated with treatment tolerance, leading to resistant clonal derivatives and disease progression (*Brown et al., 2014*). Previous studies suggested that COMPASS complex inactivation suppressed tumor suppressor genes transcription and accelerated breast cancer tumorigenesis (*Langille et al., 2022*). Aberrant DNA methylation resulted in abnormal oncogenic signaling activation and drug resistance acquisition (*Garcia-Martinez et al., 2021*). Targeting epigenetic reprogramming might provide novel treatment options for overcoming breast cancer drug resistance.

By integrating metabolomic, transcriptomic, and epigenomic data, we identified metabolic alterations, especially cysteine-associated pathways, accompanied with the epigenetic changes during the development of primary trastuzumab resistance of HER2-positive breast cancer (*Figure 1A*). Based on these discoveries, targeting certain metabolism-related proteins, epigenetic modification, or nutrition could all be possible ways to overcome trastuzumab resistance. In all, our study provided novel insights and potential solutions on primary trastuzumab resistance of HER2-positive breast cancer.

## Methods

### Patient cohort and blood samples collection

One breast cancer patient cohort from Jiangsu Province Hospital was embedded in this study, which consists of 26 primary trastuzumab-resistant patients defined as disease recurrence during or after (≤12 mo) trastuzumab adjuvant treatment and 26 trastuzumab-sensitive patients that benefited from trastuzumab adjuvant treatment more than 12 mo (*Wong et al., 2011*).

All blood samples were collected before the first trastuzumab treatment cycle. Sample collection was under the approval of the Ethics Committee and the Institutional Review Board of Jiangsu Province Hospital. Each patient provided written informed consent for sample and data use.

### Cell culture

Human breast cancer cell lines JIMT1, SKBR3, and human breast epithelial cell line MCF10A were obtained from American Type Culture Collection (ATCC). All cell types underwent cell line authentication

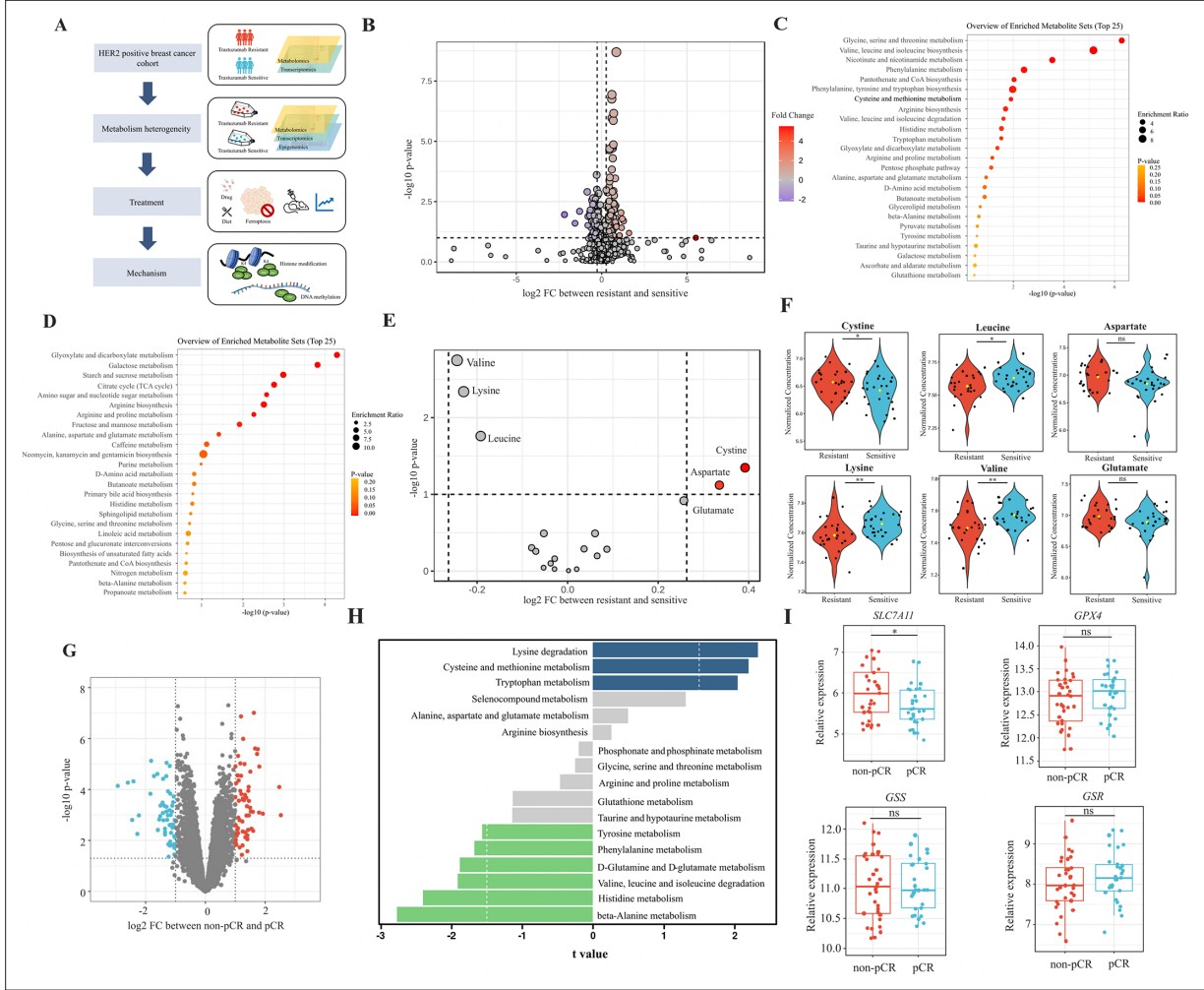

**Figure 1.** Analyses of amino acids metabolic patterns in trastuzumab primary-resistant and sensitive human epidermal growth factor receptor 2 (HER2)-positive breast cancer patients. (**A**) Workflow of analyses performed in this study. (**B**) Volcano plot of different circulating metabolites in the comparison between trastuzumab primary-resistant and sensitive patients. (**C**) Enrichment analysis of circulating metabolites downregulated in trastuzumab primary-resistant patients. (**D**) Enrichment analysis of circulating metabolites upregulated in trastuzumab primary-resistant patients. (**E, F**) Volcano plot (**E**) and violin plots (**F**) of different circulating protein-construction amino acids in the comparison between trastuzumab primary-resistant and sensitive patients. (**G**) Volcano plot of different genes between non-pathological complete response (pCR) and pCR patients in trastuzumab-based neoadjuvant treatment (I-SPY2, GSE181574). (**H**) Amino acids metabolic pathway analysis of non-pCR and pCR patients. (**I**) Relations between cysteine metabolic genes and trastuzumab treatment outcomes. Significances were determined by two-tailed unpaired t-test (**F, I**). ns, p≥0.05; *p<0.05; **p<0.01; ***p<0.001.

The online version of this article includes the following figure supplement(s) for figure 1:

**Figure supplement 1.** Quality control and differentiation analysis of plasma metabolites in trastuzumab-sensitive and primary-resistant human epidermal growth factor receptor 2 (HER2)-positive breast cancer patients.

based on short tandem repeats (STR) and were tested negative for mycoplasma contamination. No cell lines from the list of commonly misidentified cell lines maintained by the International Cell Line Authentication Committee were used in this study.

JIMT1 and SKBR3 cell lines were cultured in high-glucose Dulbecco's modified eagle medium (DMEM; Gibco, 11965092) with 10% fetal bovine serum (FBS; Gibco, 16140071) and 1% penicillin streptomycin (Gibco, 15140122) at 37°C with 5% $CO_2$. MCF10A was cultured in mammary epithelial cell growth medium (MEGM) supplemented with bovine pituitary extract, human epidermal growth factor, hydrocortisone, gentamicin sulfate-amphotericin, and insulin (Lonza, CC-3150) with 100 ng/ml cholera toxin (Sigma, C8052) at 37°C with 5% $CO_2$. Cell culture with cysteine starvation was performed by cystine/cysteine-deficient DMEM (BIOTREE) and 1% penicillin streptomycin at 37°C with 5% $CO_2$.

## DNA constructs, plasmids, and transfection

Short interfering RNA (siRNA) targeting *ASH2L*, *SLC7A11*, and *GPX4* were presented in *Supplementary file 1, table 1*. The design of small guide RNA (sgRNA) was based on the relevant methylated region and conducted via the online tool CRISPick (*Doench et al., 2016*; *Sanson et al., 2018*). Sequences of sgRNA were indicated in *Supplementary file 1, table 2*. dCas9-DNMT3A was obtained from Addgene (#100090) (*O'Geen et al., 2017*).

Transfection was conducted by using Lipofectamine 3000 (Invitrogen, L3000015) according to the manufacturer's protocol. In brief, cells were cultured in a six-well plate at 70–90% confluency. For RNA interference, 2500 ng siRNA and 3.75 μl Lipofectamine 3000 reagents were mixed and then added in each well. For CRISPR-based epigenetic editing, dCas9-DNMT3A plasmid and sgRNA were first mixed in 1:2 ratio, then mixed with 3.75 μl Lipofectamine 3000 reagent and 5 μl P3000 reagent in each well.

## Protein extraction and western blots

Cells were collected and then lysed in SDS lysis buffer with phenylmethyl-sulfonyl fluoride (PMSF; Servicebio, G2008) and phosphatase inhibitor cocktail (Servicebio, G2007) on ice. Proteins were separated in SDS-PAGE and transferred to PVDF membranes (Millipore, IPVH00010). After blocking, membranes were incubated with specific primary and secondary antibodies and then visualized with the imaging system and ECL chemiluminescence kit (Vazyme, E422-01). Primary antibodies included SLC7A11 (CST, 12691 S), GPX4 (CST, 52455 S), glutathione synthetase (Abcam, ab124811), glutathione reductase (Abcam, ab124995), ASH2L (CST, 5019T), H3K4me3 (Abcam, ab213224), GAPDH (proteintech, HRP-60004), and H3 (Abcam, ab1791). The secondary antibody was horseradish peroxidase (HRP) conjugated goat anti-rabbit IgG(H+L) (ImmunoWay, RS0002).

## GSH/GSSG ratio assay

GSH/GSSG ratio was measured with a GSH and GSSG assay kit (Beyotime, S0053). Cells were lysed with protein-removing reagent solution and several freeze-thaw cycles. After centrifugation, supernatants were collected and incubated with DTNB and NADPH. The absorbance at 412 nm was detected to measure the GSH/GSSG ratio with standard curve.

## C11 BODIPY (581/591) assay

Cells were collected and incubated with 5 μM C11 BODIPY (581/591) (Invitrogen, D3861) for 30 min at 37 °C. The oxidized and reduced states were detected by flow cytometry, with excitation and emission wavelengths at 488/510 and 581/591 nm, respectively.

## DCFH-DA assay

Cells were collected and incubated with 10 μM DCFH-DA (MCE, HY-D0940) for 30 min at 37 °C. The oxidated state was detected with a laser excitation wavelength at 488 nm.

## Cystine uptake assay

The cystine uptake ability was measured by using the cystine uptake assay kit (DOJINDO, UP05). Cells were incubated with cystine analog solution and fluorescent probe. The fluorescence intensity was measured with excitation and emission wavelengths at 490 and 535 nm.

## Quantification of intracellular cysteine

The concentration of intracellular cysteine was detected by using a cysteine quantification kit (Solarbio, BC0185). Cells were lysed and incubated with solution containing phosphotungstic acid. The absorbance at 600 nm was detected to measure intracellular cysteine levels.

## ChIP

Chromatin extraction and the following immune precipitation were performed with chromatin extraction kit (Abcam, ab117152) and ChIP kit (Abcam, ab117138) according to the manufacturer's protocols. Cells were harvested and treated with 1% formaldehyde first, then sonication was applied on cross-linked chromatin to generate 200–1000 bp fragments. Chromatin fractions enriched with

certain histone modifications were captured by antibodies (H3K27me3 (CST, C36B11), H3K4me3 (Abcam, ab213224) and IgG) and released by protease digestion.

RT-qPCR was applied to quantify the enrichment of specific types of histone modification at certain chromatin regions with ChamQ SYBR qPCR Master Mix (Vazyme, Q311). While VAHTS Universal DNA Library Prep Kit for Illumina (Vazyme, ND607) and Illumina Novaseq 6000 platform were applied for sequencing library preparation and the following sequencing. All primers used for ChIP-qPCR were presented in *Supplementary file 1, table 3*.

## MeDIP

Methylated DNA immune precipitation was performed with MeDIP kit (Abcam, ab117135) according to manufacturer's protocols followed by chromatin extraction as ChIP protocol. After cross-linking and sonication, methylated chromatin fractions were enriched by anti-5-Methylcytosine (5-mC) antibody (CST, 28692). RT-qPCR was applied to quantify the enrichment of 5-mC at certain chromatin regions. All primers used for MeDIP-qPCR were presented in *Supplementary file 1, table 3*.

## Dot blots

Total genomic DNA was extracted by using a nucleic DNA isolation kit (Vazyme, DC112). Denaturized DNA was spotted on a nitrocellulose membrane, and blocked with 5% skim milk in TBST (20 mM Tris, 150 mM NaCl, and 0.1% Tween-20). Anti-5-mC primary antibody and HRP-conjugated secondary antibody together with ECL chemiluminescence kit were applied for total methylated cytosine visualization.

## Mice xenograft model construction

Six-week-old female BALB/c nude mice were obtained from Beijing Vital River Laboratory Animal Technology, housed in a 12 hr light-dark lighting condition with 20–22°C temperature and 60 ± 10% humidity. The animal diet and drinking water followed the national standard for experimental animal feed. The diet underwent Cobalt-60 irradiation sterilization, and the drinking water followed purification and sterilization.

All animal experiments were conducted according to the review and approval of the Institutional Animal Care and Use Committee in Nanjing Medical University (IACUC-2204057). $3 \times 10^6$ JIMT1 cells were injected subcutaneously into the mammary fat pad region to generate xenograft models.

The mice were divided into six groups: (*Giaquinto et al., 2022*) DMSO treatment; (*Harbeck et al., 2019*) trastuzumab (Roche) treatment (10 mg/kg, injected intraperitoneally twice a week); (*Swain et al., 2023*) erastin (HY-15763) treatment (40 mg/kg, injected intraperitoneally every other day); (*Marra et al., 2024*) cysteine starvation; (*Cossetti et al., 2015*) trastuzumab treatment plus cysteine starvation; (*Xu et al., 2017*) trastuzumab plus erastin treatment. Cysteine starvation was performed by utilizing cystine/cysteine deficient diet (Xietong Bio). Treatment was initiated when tumor volume reached 50 mm³. Tumor sizes and mice weights were measured three times a week. Tumor volumes were calculated as $0.5 \times L \times W^2$ (L: the longest dimension, W: the perpendicular dimension). The tumor growth was compared by the paired individual fold change of tumor volumes.

## Tumor and spleen cell composition analysis

Tumor samples were digested in serum-free RPMI with 10 mg/ml collagenase I (Sigma, C0130) and 1 mg/mg DNase I (Roche, 10104159001) for 30–60 min at 37°C. Spleen samples were gently crushed. Both of them were filtered in 70 µm strainers, followed by red blood cell removement with RBC lysis buffer (eBioscience, 00-4333-57), and then resuspended in flow cytometry staining buffer. Fc receptor was blocked with mouse FcR blocking reagent (Miltenyi, 130-092-575).

Fluorescent antibodies including CD45 (clone 30-F11, eBioscience, 48-0451-82), CD3 (clone 145–2 C11, BioLegend, 100353), CD49b (clone DX5, BioLegend, 108907), CD11b (clone M1/70, eBioscience, 56-0112-82), CD27 (clone LG.3A10, BioLegend, 124216), CD335 (NKp46) (clone 29A1.4, eBioscience, 17-3351-82), and granzyme B (clone NGZB, eBioscience, 11-8898-82) together with fixable viability dye (eBioscience, 65-0865-14) were chosen for cell staining. CytoFLEX (Beckman Coulter) and FlowJo software were applied for fluorescence measurement and analysis.

## Immunohistochemical staining

After formalin fixation and paraffin embedding, tissue sections were cut and mounted on slides followed by deparaffinizing, rehydrating, and antigen retrieval. Samples were then stained with

antibodies including SLC7A11 (Abcam, ab307601), GPX4 (Abcam, ab125066), 4-Hydroxynonenal (4-HNE) (Abcam, ab48506), Malondialdehyde (MDA) (Abcam, ab243066), and caspase-3 (Abcam, ab32150).

## Differential transcriptome analyses

The human hg38 genome were utilized for RNA sequencing read alignment. Differential expression analyses were measured by the DESeq2 package in R. Significant differential expression was defined with a threshold of p adj ≤0.05 and |log2 fold change|≥0.5.

## Differential metabolome analyses

Metabolomics data were analyzed by the MetaboAnalystR package in R (*Ewald et al., 2024*). Pathway enrichment analyses were performed according to KEGG databases. Significant difference was defined with a threshold of p adj ≤0.05 and |log2 fold change|≥0.5.

## Differential epigenome analyses

Sequencing reads of ChIP and whole-genome bisulfite sequencing (WGBS) were aligned to the human hg38 genome. Differential methylation analyses were performed by deepTools (version 3.5.5) (*Ramírez et al., 2016*). Differentially methylated regions (DMR) were identified by Bismark (version 0.24.2) (*Krueger and Andrews, 2011*). Significant difference was defined with a threshold of p adj ≤0.05 and |log2 fold change|≥0.5.

## Genomics analyses

The human hg38 genome was utilized to align WGS sequencing reads. Germline single nucleotide polymorphisms (SNP) and indels were identified by GATK (v 4.5.0.0). The annotation of germline mutations was obtained by utilizing ANNOVAR.

## Statistical analyses

All statistical analyses were performed by utilizing GraphPad Prism (version 10.1.2) and R (version 4.3.3). All data were represented as the mean ± standard error (SE) or the mean ± SEM from a minimum of three independent experiments. Two-tailed Student's t-test was applied to determine statistical significance in comparisons between two groups and ANOVA test in multiple groups. The threshold for statistical significance is $p<0.05$, and was marked as *$p<0.05$, **$p<0.01$, ***$p<0.001$.

# Results

## Different amino acids metabolic patterns between trastuzumab-sensitive and primary-resistant HER2-positive breast cancer patients

To investigate metabolic patterns of HER2-positive breast cancer patients with different trastuzumab response, we obtained and analyzed the plasma metabolites data of both trastuzumab-sensitive and primary-resistant patients (26/26) with good interclass correlations (*Figure 1—figure supplement 1A–C*). Of them, 203 metabolites (130 upregulated and 73 downregulated in primary-resistant groups) were identified with different abundances among two patient groups (*Figure 1B* and *Figure 1—figure supplement 1D*). KEGG-based pathway analyses suggested the major enrichment in amino acid metabolism. For instance, glycine, serine, and threonine metabolism, valine, leucine, and isoleucine biosynthesis, as well as cysteine and methionine metabolism, were reduced in primary-resistant patients; while arginine, proline, alanine, aspartate, and glutamate metabolism were enhanced (*Figure 1C and D*).

By comparing the abundance of 20 common protein-construction amino acids, we realized that cystine and aspartate were higher, while valine, lysine, and leucine were lower in primary-resistant groups. However, glutamate kept stable in both groups (*Figure 1E and F*). Non-protein amino acids also displayed differences between these two groups. 2-aminoisobutyric acid and 4-aminobutyric acid (GABA) were lower in resistant groups, while citrulline, kynurenine, ornithine, and taurine levels were similar (*Figure 1—figure supplement 1E*).

We then analyzed transcriptomic data of pre-treatment breast cancer tumors in the I-SPY2 trial (*Clark et al., 2021*) to further explore metabolic gene alterations, in which HER2-positive breast

cancer patients received trastuzumab plus paclitaxel, trastuzumab combined with pertuzumab, and paclitaxel or T-DM1 plus paclitaxel during neoadjuvant settings. By comparing patients with different responses to neoadjuvant trastuzumab-based regimens (pathological complete response [pCR] and non-pCR), we found 1542 genes upregulated and 1350 genes downregulated in non-pCR groups (*Figure 1—figure supplement 1F* and *Figure 1G*). Based on KEGG datasets, amino acid metabolism was found dysregulated in patients with poor responses to trastuzumab, including aberrant lysine degradation, cysteine and methionine metabolism, beta-alanine metabolism, and histidine metabolism (*Figure 1H*).

As cystine was one of the most significant amino acids in both metabolomic and transcriptomic analyses, we investigated relations between cystine/cysteine metabolic key genes and trastuzumab treatment outcomes. Patients with non-pCR featured higher expression of *SLC7A11*, while no differences were found in *GPX4*, *GSS,* and *GSR* expression (*Figure 1I*), indicating *SLC7A11* might become a potential target for trastuzumab primary-resistant patients.

In general, trastuzumab primary-resistant HER2-positive breast cancer patients featured distinct amino acid metabolic patterns, especially increased plasma cystine levels and higher expression of *SLC7A11*.

## Distinct cysteine metabolism in trastuzumab primary-resistant HER2-positive breast cancer

To comprehensively compare the metabolic heterogeneity between trastuzumab-sensitive and primary-resistant HER2-positive breast cancer, we obtained and analyzed both metabolomic and transcriptomic data of trastuzumab-sensitive SKBR3 and trastuzumab-resistant JIMT1 cell lines.

Significant differences were observed in metabolic data with good quality (*Figure 2—figure supplement 1A and B*). Specifically, 181 upregulated and 235 downregulated hydrophilic metabolites accompany 546 upregulated and 412 downregulated lipophilic metabolites were found in JIMT1 cells compared to SKBR3 cells (*Figure 2A* and *Figure 2—figure supplement 1C*). For hydrophilic metabolites, enrichment analyses revealed that glycine, serine, and threonine metabolism were upregulated in JIMT1, while cysteine and methionine metabolism were downregulated (*Figure 2B and C*). Enrichment analyses based on lipophilic metabolites showed that linoleic acid metabolism, fatty transportation, and glycerophospholipid catabolism were upregulated in JIMT1, but glycerolipid metabolism, phospholipid biosynthesis, and retinoid metabolism were downregulated (*Figure 2—figure supplement 2A and B*).

Differential gene expression could also be uncovered from transcriptomic data. 4235 downregulated genes and 4923 upregulated genes were identified in the comparison between JIMT1 and SKBR3 cells (*Figure 2—figure supplement 3A*). Based on KEGG datasets, activities of cysteine and glutathione metabolism pathways were found dysregulated in JIMT1 cells (*Figure 2D*). More importantly, the joint pathway analyses combining metabolomic and transcriptomic data indicated that JIMT1 featured aberrant glutathione metabolism (*Figure 2E*).

The extracellular cystine is mainly imported by SLC7All for subsequent protein synthesis, glutathione (GSH) production, and oxidative catabolism (*Figure 2F*). Both transcriptomic and protein evidence revealed the upregulation of SLC7A11 and glutathione synthetase (GSS) as well as the downregulation of GPX4 and glutathione reductase (GSR) in JIMT1 cells (*Figure 2—figure supplement 3B* and *Figure 2G*), together with the characteristics that JIMT1 cells showed higher cysteine uptake ability and cytoplasmic cysteine level but lower glutathione/glutathione disulfide (GSH/GSSG) ratio than SKBR3 cells (*Figure 2H–J*), indicating the probably increased transsulfuration activity and decreased γ-glutamyl-peptides synthesis activity in JIMT1 cells (*Figure 2—figure supplement 3C*). These results suggested that HER2-positive breast cancer with primary trastuzumab resistance might prefer to consume more cystine and feature intracellular cysteine metabolic reprogramming, showing high consistency with metabolic characteristics of relevant breast cancer patients and may associate with trastuzumab treatment response.

## Stimulate ferroptosis sensitivity of trastuzumab primary-resistant HER2-positive breast cancer

Since cysteine metabolism plays a key role in maintaining cellular oxidative-reductive balance, disruption of which may turn breast cancer cells into a more sensitive state to oxidative damage, including

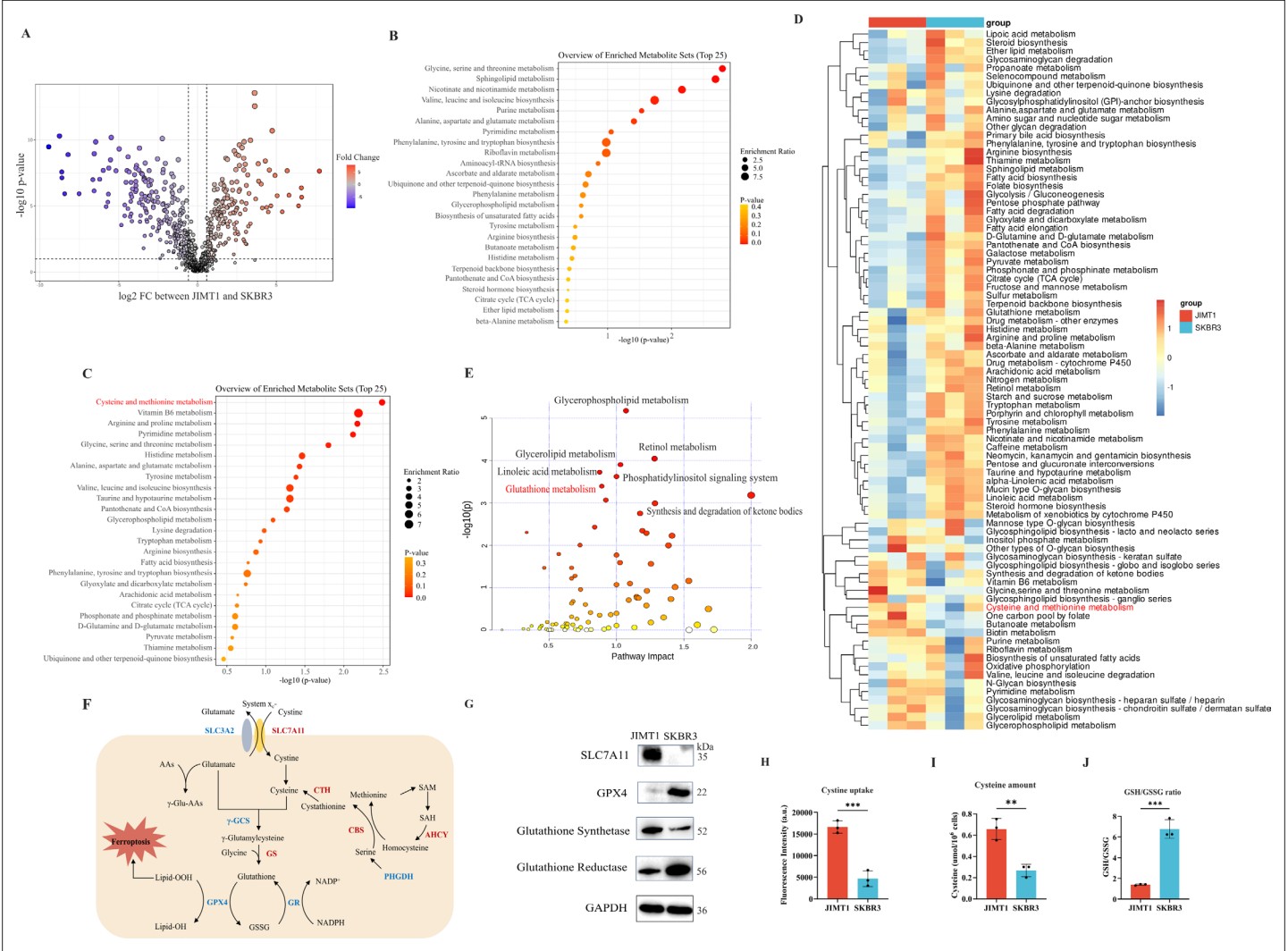

**Figure 2.** Heterogeneity of cysteine metabolism characteristics in human epidermal growth factor receptor 2 (HER2)-positive breast cancer with different trastuzumab responses. (**A**) Volcano plot of hydrophilic metabolites in JIMT1 and SKBR3. (**B**) Upregulated metabolic processes in JIMT1 based on enrichment analysis of metabolites. (**C**) Downregulated metabolic processes in JIMT1 based on enrichment analysis of metabolites. (**D**) Different metabolic activities between JIMT1 and SKBR3 based on transcriptomics data. (**E**) Joint pathway analysis based on metabolomics and transcriptomics data. (**F**) Map of essential genes involved with cystine/cysteine metabolism. Upregulated genes in JIMT1 are highlighted in red and downregulated genes in JIMT1 are highlighted in blue. (**G**) Expression of essential proteins participating in cysteine and glutathione metabolism. (**H**) The comparison of cystine uptake ability between JIMT1 and SKBR3. (**I**) The comparison of intracellular cysteine abundance between JIMT1 and SKBR3. (**J**) The comparison of GSH:GSSG ratio between JIMT1 and SKBR3. Significances were determined by two-tailed unpaired t-test (**H–J**). ns, p≥0.05; *p<0.05; **p<0.01; ***p<0.001.

The online version of this article includes the following source data and figure supplement(s) for figure 2:

**Source data 1.** PDF file containing original western blots for *Figure 2G*, indicating relevant bands.

**Source data 2.** Original files for western blot analysis displayed in *Figure 2G*.

**Source data 3.** Raw data files for *Figure 2H-J*.

**Figure supplement 1.** Quality control and differentiation analysis of metabolomic data in JIMT1 and SKBR3.

**Figure supplement 2.** Enrichment analysis of different lipophilic metabolites in JIMT1 and SKBR3.

**Figure supplement 3.** Analysis of transcriptomic data in JIMT1 and SKBR3.

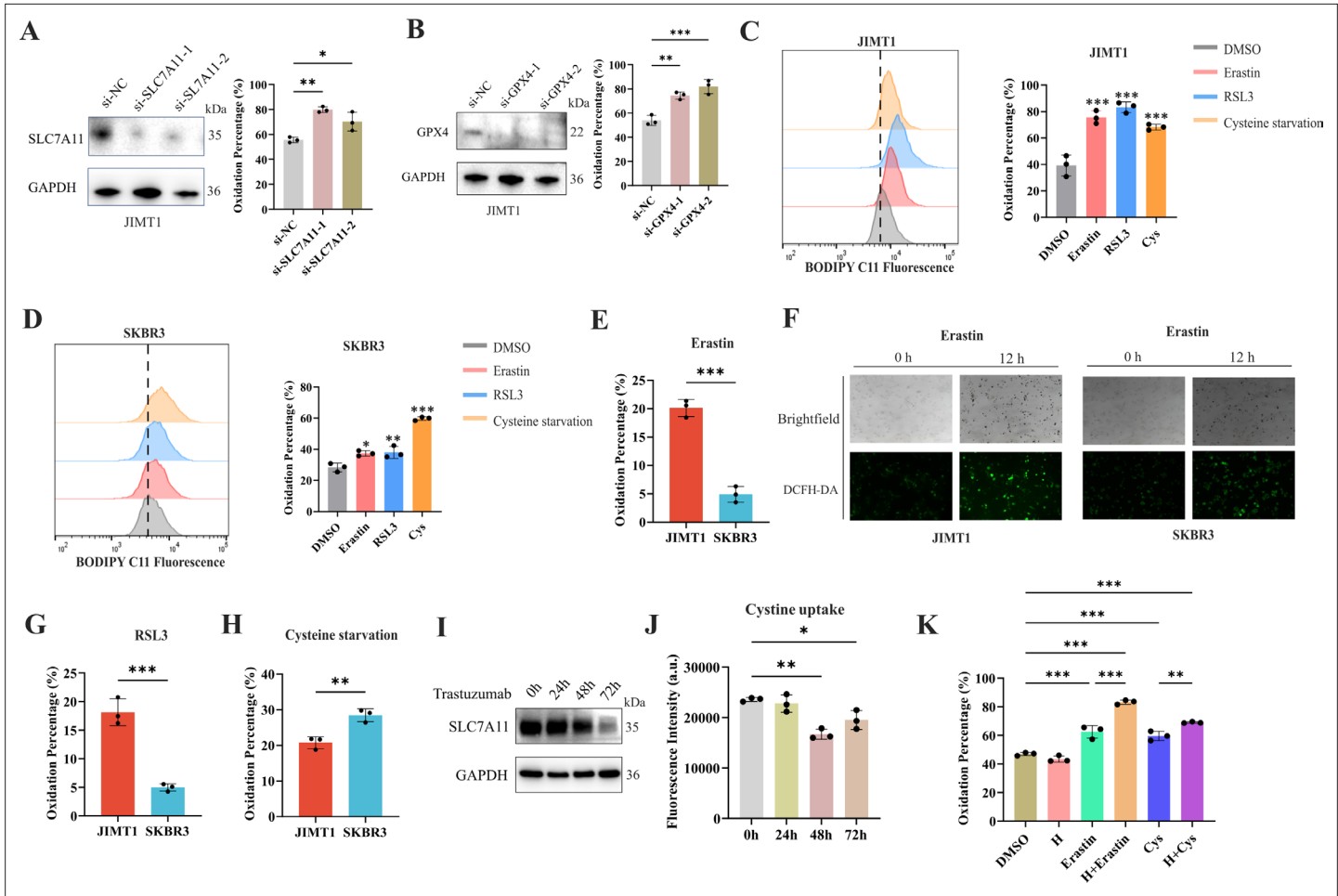

**Figure 3.** Ferroptosis sensitivity of human epidermal growth factor receptor 2 (HER2)-positive breast cancer with different trastuzumab response. (**A, B**) Knockdown of *SLC7A11* (**A**) and *GPX4* (**B**) induced increased lipid peroxidation (BODIPY-C11) in JIMT1. (**C, D**) Treatment with erastin (10 μm), RSL3 (1 μm), and cysteine starvation for 12 hr induced lipid peroxidation (BODIPY-C11) in JIMT1 (**C**) and SKBR3 (**D**). (**E, F**) JIMT1 and SKBR3 featured different lipid peroxidation sensitivity to erastin by measuring BODIPY-C11 (**E**) and DCFH-DA (**F**). (**G, H**) JIMT1 and SKBR3 featured different lipid peroxidation sensitivity to RSL3 (**G**) and cysteine starvation (**H**). (**I**) SLC7A11 expression under different trastuzumab treatment time. (**J**) Cystine uptake ability under different trastuzumab treatment time. (**K**) The combination of erastin or cysteine starvation and trastuzumab increased lipid peroxidation in JIMT1. H, trastuzumab; Cys, cysteine starvation. Significances were determined by one-way ANOVA (**A–D, J, K**) and two-tailed unpaired t-test (**E, G, H**). ns, p≥0.05; *p<0.05; **p<0.01; ***p<0.001.

The online version of this article includes the following source data and figure supplement(s) for figure 3:

**Source data 1.** PDF file containing original western blots for *Figure 3A, B and C*, indicating relevant bands.

**Source data 2.** Original files for western blot analysis displayed in *Figure 3A, B and C*.

**Source data 3.** Raw data files for *Figure 3A–E, G, H, J and K*.

**Figure supplement 1.** Knockdown of genes and targeting cysteine metabolism in JIMT1 and SKBR3.

**Figure supplement 1—source data 1.** PDF file containing original western blots for *Figure 3—figure supplement 1A and B*, indicating relevant bands.

**Figure supplement 1—source data 2.** Original files for western blot analysis displayed in *Figure 3—figure supplement 1A and B*.

**Figure supplement 1—source data 3.** Raw data files for *Figure 3—figure supplement 1A–E, G and H*.

ferroptosis (*Dixon and Olzmann, 2024*; *Ye et al., 2024*). We first downregulated the expression of *SLC7A11* and *GPX4* in JIMT1 by siRNAs, and the lipid peroxidation of ferroptosis was found to increase by measuring BODIPY-C11 (*Figure 3A and B*). Similar results were also detected in SKBR3 (*Figure 3—figure supplement 1A and B*). We then detected the sensitivity of JIMT1 and SKBR3 to SLC7A11 inhibitor (erastin) and GPX4 inhibitor (RSL3) and found JIMT1 was more sensitive to these two inhibitors (*Figure 3—figure supplement 1C and D*). The deprivation of cysteine could suppress

the growth of both JIMT1 and SKBR3, but JIMT1 showed more tolerance to this nutritional starvation (*Figure 3—figure supplement 1E*).

12 hr treatment of SLC7A11 inhibitor (erastin), GPX4 inhibitor (RSL3), and even cysteine starvation were all able to induce ferroptosis in both JIMT1 and SKBR3 cells (*Figure 3C and D*). However, JIMT1 cells indicated more rapid accumulation of lipid peroxidation under the treatment of erastin, which could be detected by both increased BODIPY-C11 and DCFH-DA (*Figure 3E and F*). Compared with SKBR3, JIMT1 cells were also more sensitive to RSL3 but more durable to cysteine starvation (*Figure 3G and H*), which might be explained by their stronger transsulfuration activities and decreased γ-glutamyl-peptides synthesis (*Figure 3—figure supplement 1F*). The utilization of ferroptosis inhibitor Fer-1 could suppress the lipid peroxidation resulting from erastin, RSL3, and cysteine starvation in both JIMT1 and SKBR3, which also indicated the role of cysteine metabolism in modulating HER2-positive breast cancer ferroptosis (*Figure 3—figure supplement 1G and H*).

To understand whether trastuzumab would modulate the ferroptosis sensitivity, we treated JIMT1 cells with trastuzumab for different durations. We found that with a prolonged exposure of trastuzumab for 72 hr, the total amount of SLC7A11 in JIMT1 was decreased (*Figure 3I*). Correspondingly, cystine uptake ability was suppressed after trastuzumab treatment for 48 and 72 hr (*Figure 3J*). In addition, cells that underwent trastuzumab treatment combined with erastin or cysteine starvation could demonstrate stronger lipid peroxidation and enhanced ferroptosis (*Figure 3K*).

## Restraining cysteine metabolism inhibited tumor growth in vivo

To further investigate the effects of SLC7A11 inhibition or cysteine limitation on trastuzumab primary-resistant HER2-positive breast cancer, JIMT1 xenograft mouse models were generated and then treated with different strategies, including DMSO, trastuzumab, erastin, cysteine starvation, trastuzumab plus erastin, and trastuzumab plus cysteine starvation (*Figure 4A*).

Of them, trastuzumab plus erastin and trastuzumab plus cysteine starvation could effectively inhibit tumor growth, while cysteine starvation and erastin monotherapy only showed limited effects on tumor progression, demonstrating the better treatment responses of the combination strategy (*Figure 4B and C* and *Figure 4—figure supplement 1A*). In addition, almost no weight differences were observed among all groups during treatment, partially indicating the safety of such cysteine limitation treatment (*Figure 4D*).

The immune cell compositions in tumor and spleen of each group were measured to analyze the immune response induced by different treatment strategies (*Figure 4—figure supplement 1B and C*). The infiltration of NK cells (CD3-CD45+CD49b+) in tumors was increased only upon the combination of cysteine deprivation and trastuzumab (*Figure 4E*). Nearly no differences were found in tumor-infiltrating NK cell activation (NKp46), toxic function (Granzyme B), and maturation status (CD11b, CD27) among all these groups (*Figure 4F–H*). While similar results were also observed in spleen NK cells (*Figure 4—figure supplement 1D–G*).

These results suggested a better effect of cysteine starvation than erastin in NK cell recruitment and activation, resulting in a more powerful synergy with trastuzumab to increase the abundance of tumor-infiltrating NK cells and show better inhibition of trastuzumab primary-resistant breast cancer. As the anti-tumor effects of trastuzumab mainly relies on antibody-dependent cellular cytotoxicity (ADCC) effects mediated by NK cells (*Musolino et al., 2022*), this combination therapy might provide more effective approaches for HER2-positive breast cancer treatment.

We then analyzed the biomarkers of cysteine metabolism and cell death pathways in tumor samples of different treatment groups (*Figure 4I*). Erastin and cysteine deprivation monotherapy, or combined with trastuzumab, could upregulate SLC7A11 and GPX4 expression, as well as the abundance of MDA and 4-HNE. Ferroptosis might increase endoplasmic reticulum stress and activate p53 upregulated modulator of apoptosis (PUMA) (*Lee et al., 2020*). The combination of trastuzumab and erastin or trastuzumab and cysteine starvation might also modify apoptosis sensitivity. Apoptosis biomarker caspase-3 was found increasing in these synergic treatment groups (*Figure 4—figure supplement 1H*). These results suggested that targeting cysteine metabolism could induce ferroptosis in trastuzumab primary-resistant HER2-positive breast cancer and might have a synergic effect with trastuzumab to inhibit breast cancer progression.

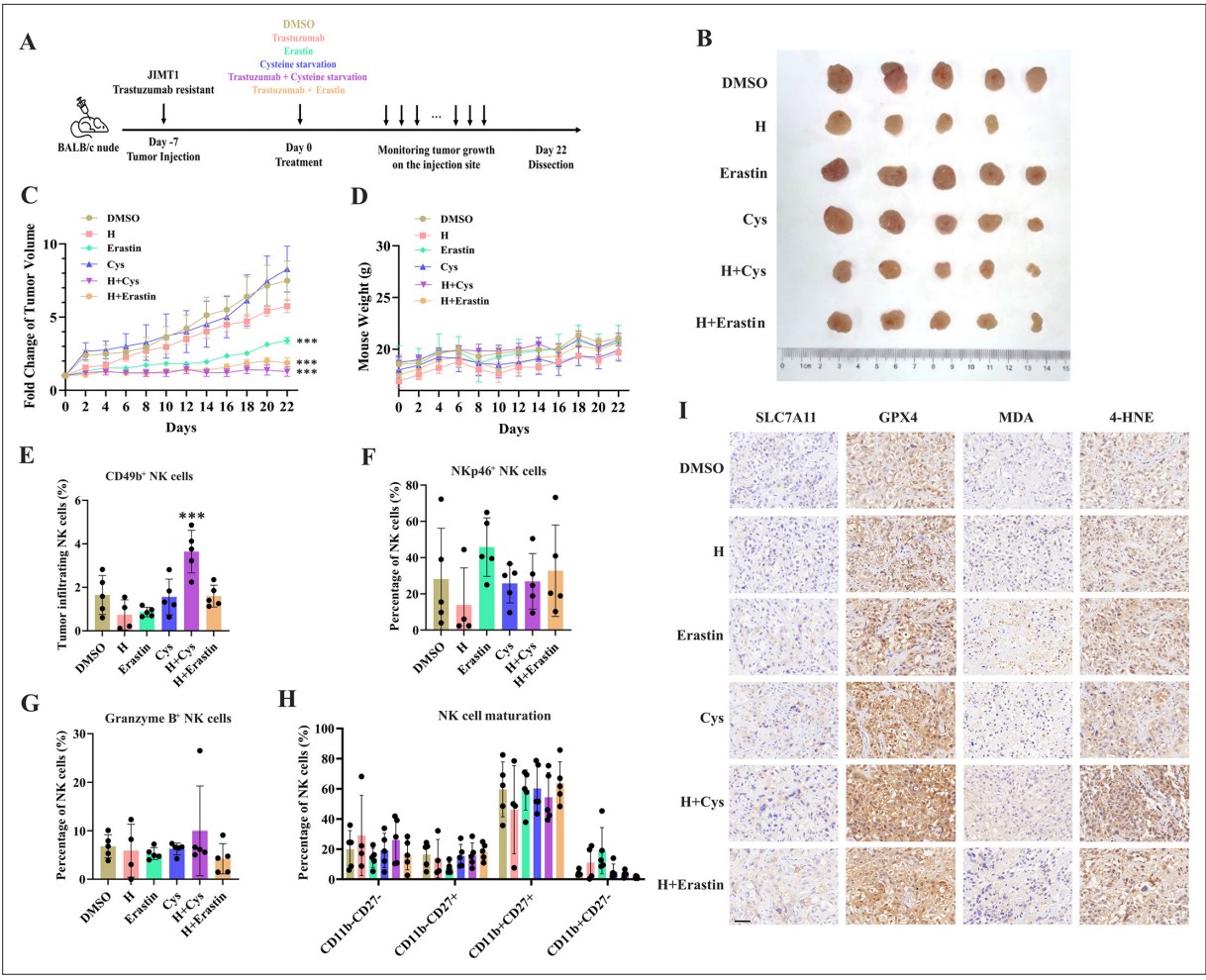

**Figure 4.** Targeting cysteine metabolism synergizes with trastuzumab to induce ferroptosis. (**A**) Schematic outline of treatment on cysteine metabolism in trastuzumab-resistant breast cancer. JIMT1 cells were injected into the mammary fat pads of BALB/c nude mice at day 7. Treatment with DMSO, erastin, cysteine starvation, trastuzumab combined with or without erastin and cysteine starvation started at day 0 for 21 d. Tumor volume and mice weight were measured. The tumor growth was compared by paired individual fold change of tumor volumes. (**B**) Tumor of different treatment groups. (**C**) Fold change of tumor volume in different treatment groups. (**D**) Mice weight of different treatment groups. (**E–H**) Abundance of CD49b+ (**E**), NKp46+ (**F**), Granzyme B+ (**G**), and different development status (**H**) tumor-infiltrating NK cells. (**I**) Representing immunohistochemistry images of SLC7A11, GPX4, MDA, and 4-HNE in different treatment groups. Scale bar, 50 μm. H trastuzumab; Cys, cysteine starvation. Significances were determined by two-way ANOVA (**C, D**) and one-way ANOVA (**E–H**). ns, $p \geq 0.05$; *$p < 0.05$; **$p < 0.01$; ***$p < 0.001$.

The online version of this article includes the following source data and figure supplement(s) for figure 4:

**Source data 1.** Raw data files for *Figure 4C–H*.

**Figure supplement 1.** Detection of immune cell compositions in tumor and spleen samples.

**Figure supplement 1—source data 1.** Raw data files for *Figure 4—figure supplement 1A and D–G*.

## Aberrant promoter H3K4me3 regulates SLC7A11 gene expression and cysteine metabolism

Abnormal changes of histone modification play critical roles in tumor progression, including drug resistance (*Jin and Jeong, 2023*). Of them, altered methylations on promoter histone H3K4 and H3K27 show the most important effects on gene expression (*Millán-Zambrano et al., 2022*).

By comparing the ChIP-seq data, we detected 3934 and 3925 H3K4me3 peaks were enriched in JIMT1 and SKBR3, respectively. These altered peaks primarily located among the promoter regions of genes (*Figure 5A* and *Figure 5—figure supplement 1A*), which have close relationships with biological processes such as nucleic acid transcription, protein modification, mRNA processing, and ribosome processing (*Figure 5—figure supplement 2A*).

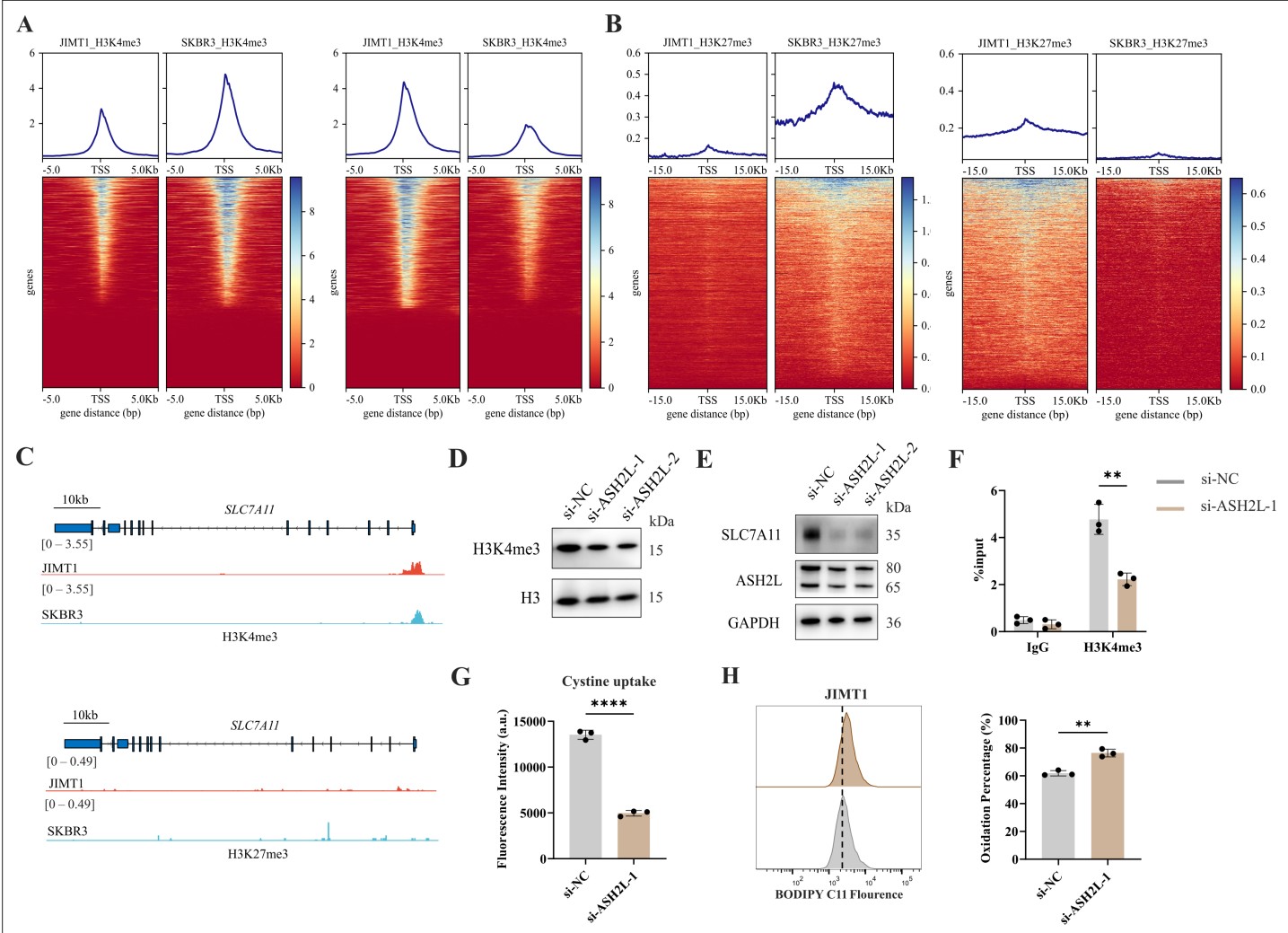

**Figure 5.** Increased H3K4me3 modifies *SLC7A11* transcription and regulates cystine uptake. (**A, B**) Alterations of H3K4me3 (**A**) and H3K27me3 (**B**) peaks between JIMT1 and SKBR3. (**C**) Different abundance of H3K4me3 and H3K27me3 at *SLC7A11* promoter regions in JIMT1 and SKBR3. (**D**) Decreased ASH2L inhibited total H3K4me3 expression. (**E**) Decreased ASH2L inhibited SLC7A11 expression. (**F**) Decreased ASH2L suppressed H3K4me3 expression at SLC7A11 promoter regions. (**G**) Decreased ASH2L suppressed cystine uptake ability. (**H**) Decreased ASH2L induced lipid peroxidation and ferroptosis. Significances were determined by two-tailed unpaired t-test (**F–H**). ns, p≥0.05; *p<0.05; **p<0.01; ***p<0.001.

The online version of this article includes the following source data and figure supplement(s) for figure 5:

**Source data 1.** PDF file containing original western blots for *Figure 5D and E*, indicating relevant bands.

**Source data 2.** Original files for western blot analysis displayed in *Figure 5D and E*.

**Source data 3.** Raw data files for *Figure 5F–H*.

**Figure supplement 1.** Location of altered H3K4me3 and H3K27me3 peaks in JIMT1 and SKBR3.

**Figure supplement 2.** Enrichment analysis of biological processes related to H3K4me3 and H3K27me3 alterations.

We also confirmed that 7048 and 7001 H3K27me3 peaks were enriched in JIMT1 and SKBR3, respectively, mainly enriching at not only promoter regions but also distal intergenic regions, and exons (*Figure 5B* and *Figure 5—figure supplement 1B*) of genes that are involved in cell-cell adhesions, axon guidance, postsynaptic transmission, and neuron differentiation (*Figure 5—figure supplement 2B*).

Particularly, we uncovered the enrichment of H3K4me3 at the promoter regions of key cysteine transporter gene *SLC7A11* in JIMT1 cells compared with SKBR3 cells, while H3K27me3 signals were stable (*Figure 5C*), showing the consistency with its high expression in JIMT1 cells.

By inhibiting the expression of ASH2L, one crucial subunit of COMPASS complex (*Piunti and Shilat-ifard, 2016*), we downregulated not only the overall H3K4me3 level but also the signal at *SLC7A11* promoter in JIMT1 cells, and successfully decreased the expression of SLC7A11 (*Figure 5D–F*). Meanwhile, diminished downstream cysteine uptake led to increased lipid peroxidation, the key symbol of ferroptosis (*Figure 5G and H*).

These results suggested that H3K4me3 may play a more important role in modulating SLC7A11 transcription and cysteine metabolism in JIMT1 than H3K27me3. Targeting H3K4me3 could regulate SLC7A11 expression, enhancing lipid peroxidation and inducing ferroptosis.

## DNA methylation modifies SLC7A11 transcription and cysteine metabolism

Besides histone modification, DNA methylation is another crucial epigenetic modification to regulate genomic imprinting, modify transposable elements, and modulate transcription, even in tumor progression (*Mattei et al., 2022*).

In general, JIMT1 cells featured a lower abundance of total 5-mC than SKBR3 cells (*Figure 6—figure supplement 1A*). By investigating DNA methylation landscape with WGBS, we identified 329419 CG-DMRs enriched in JIMT1 and 84634 CG-DMRs enriched in SKBR3 (*Figure 6A* and *Figure 6—figure supplement 1B*). Among non-CG DMRs (CHG, CHH; in which H denotes A, C, or T), 1604 CHG-DMRs and 3727 CHH-DMRs were upregulated in JIMT1, while 2187 CHG-DMRs and 5158 CHH-DMRs were upregulated in SKBR3 (*Figure 6—figure supplement 1B*).

Across the whole genome, CG-DMRs mainly enriched in intron, intergenic regions, and exons (*Figure 6—figure supplement 1C*). Differentially methylated genes were related to several biological processes, such as neurotransmitter biosynthesis, postsynaptic transmission, cytoplasmic translation, and ribosomal activities (*Figure 6—figure supplement 2A and B*).

*SLC7A11* is located at chromosome 4q28.3 and we found that, in comparison with SKBR3, the CpG methylation at *SLC7A11* promoter region was decreased in JIMT1 (*Figure 6B*). To further verify the influence of altered promoter CG-DMR on *SLC7A11* gene transcription, CRISPR/Cas9-based genomic editing tool dCas9-DNMT3A together with *SLC7A11*-specific promoter-targeting sgRNA sets were utilized to successfully increase the 5-mC level at *SLC7A11* promoter regions (*Figure 6C*) in JIMT1 cells, which is confirmed by MeDIP-qPCR (*Figure 6D*). As a result, the expression of SLC7A11 and cysteine uptake were reduced (*Figure 6E and F*), and the following cellular lipid peroxidation and ferroptosis were stimulated in JIMT1 cells (*Figure 6G*).

These results revealed that increased DNA methylation at *SLC7A11* promoter regions in JIMT1 might contribute to SLC7A11 overexpression. Precise modulation of 5-mC could suppress cysteine uptake and result in ferroptosis, which might provide novel methods to overcome trastuzumab resistance.

## Discussion

Metabolic heterogeneity has been widely accepted as an important characteristic in many cancer types, which are closely associated with cancer progression, distant metastasis, and even drug resistance (*Faubert et al., 2020*; *Kim and DeBerardinis, 2019*), while targeting dysregulated metabolism could also be novel therapeutic strategies (*Stine et al., 2022*).

By comparing the metabolomic data of cell lines and blood samples from patients, we realized that widespread metabolic reprogramming, particularly the alternation of cysteine metabolism was an important feature of primary trastuzumab resistance, extending our understanding of anti-tumor drug resistance (*Figure 7*).

Cysteine is a crucial non-essential amino acid and mainly obtained from dietary, protein catabolism, and methionine transsulfuration (*Bonifácio et al., 2021*). Due to oxidative extracellular environment, cystine, instead of cysteine, is normally taken in by the major transporter SLC7A11 (also known as xCT) (*Koppula et al., 2021*), which also has important roles in protein synthesis, posttranslational modifications, and redox balance maintenance (*Bonifácio et al., 2021*; *Lennicke and Cochemé, 2021*).

Meanwhile, biosynthesis of GSH by GSS is one important outlet of cellular cysteine, which helps maintaining oxidation-reduction balance and protecting membrane from oxidative damage including ferroptosis with GPX4 and GSR (*Sies et al., 2024*; *Wu et al., 2024*). The downregulation of both

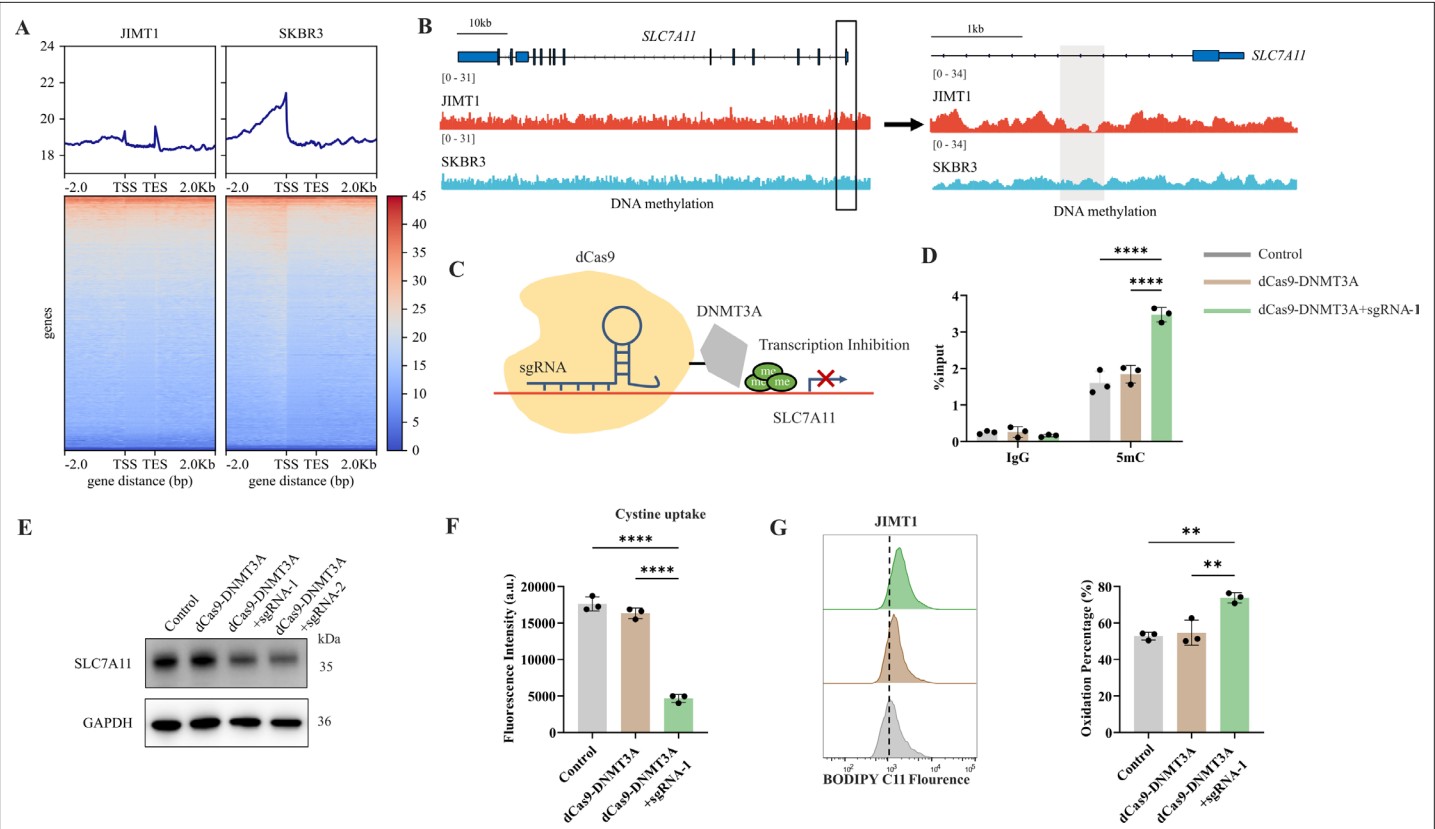

**Figure 6.** Decreased 5-mC modifications on *SLC7A11* promoter relate with enhanced cystine uptake. (**A**) Alterations of 5-mC peaks between JIMT1 and SKBR3. (**B**) Different abundance of 5-mC at *SLC7A11* promoter regions in JIMT1 and SKBR3. (**C**) Schematic outline of CRISPR-based epigenetic editing. dCas9-DNMT3A increased methylation of specific CpG islands in *SLC7A11* promoter regions and inhibited gene transcription. (**D**) dCas9-DNMT3A increased 5-mC levels at *SLC7A11* promoter regions. (**E**) dCas9-DNMT3A inhibited SLC7A11 expression. (**F**) dCas9-DNMT3A suppressed cystine uptake ability. (**G**) dCas9-DNMT3A induced lipid peroxidation and ferroptosis. Significances were determined by one-way ANOVA (**D, F, G**). ns, $p \geq 0.05$; *$p<0.05$; **$p<0.01$; ***$p<0.001$.

The online version of this article includes the following source data and figure supplement(s) for figure 6:

**Source data 1.** PDF file containing original western blots for *Figure 6E*, indicating relevant bands.

**Source data 2.** Original files for western blot analysis displayed in *Figure 6E*.

**Source data 3.** Raw data files for *Figure 6D, F and G*.

**Figure supplement 1.** Different DNA methylation status between JIMT1 and SKBR3.

**Figure supplement 1—source data 1.** PDF file containing original western blots for *Figure 6—figure supplement 1A*, indicating relevant bands.

**Figure supplement 1—source data 2.** Original files for western blot analysis displayed in *Figure 6—figure supplement 1A*.

**Figure supplement 2.** Enrichment analysis of biological processes related to DNA methylation alterations.

**Figure supplement 3.** Genomic variations in JIMT1, SKBR3, and MCF10A.

**Figure supplement 4.** SNV characteristics in JIMT1, SKBR3, and MCF10A.

**Figure supplement 5.** Density of SNP in JIMT1 (**A**), SKBR3 (**B**), and MCF10A (**C**) across the whole genome.

**Figure supplement 6.** Single nucleotide variations (SNVs) located in the exon region of *SLC7A11* in JIMT1 (**A**), SKBR3 (**B**), and MCF10A (**C**).

**Figure supplement 7.** Different histone modifications and DNA methylation in *SLC7A11* between JIMT1, SKBR3, and MCF10A.

GPX4 and GSR and the upregulation of GSS, together with the higher intake of cysteine mediated by raised SLC7A11, could stimulate GSH consumption and GSSG accumulation in JIMT1 cells, resulting in a weaker reductive cytoplasmic environment and higher sensitivity to oxidative damage, including ferroptosis.

As an important rate-limiting metabolite of the GSH biosynthesis, cysteine modulates ferroptosis through the cyst(e)ine/GSH/GPX4 axis (*Stockwell, 2022*). SLC7A11 inhibition, cysteine deprivation,

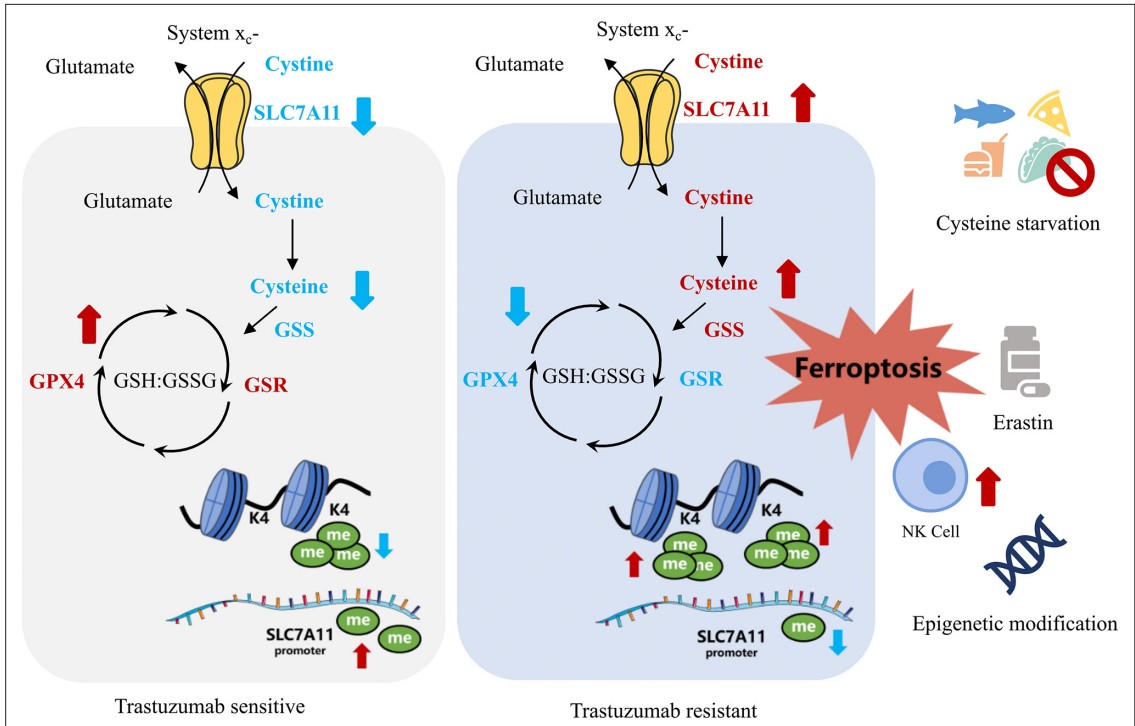

**Figure 7.** Summary of targeting cysteine metabolism in trastuzumab-resistant human epidermal growth factor receptor 2 (HER2)-positive breast cancer.

or targeting GPX4 have been reported to effectively enhance lipid peroxidation and induce ferroptosis in several cancer types (*Dixon et al., 2012*; *Badgley et al., 2020*; *Shi et al., 2021*; *Yang et al., 2023*). In our study, JIMT1 was much more sensitive to SLC7A11 and GPX4 inhibition, but tolerable to cysteine starvation, and this might result from increased methionine transsulfuration activity and γ-glutamyl-peptides synthesis, which compensated for the deficiency of extracellular cysteine (*Zhang et al., 2022*; *Kang et al., 2021*). Whether similar effects could be observed on GSS or GSR also need to be studied.

The inhibition of HER2 could result in impaired redox homeostasis and increased ROS (*Fox et al., 2020*). Tumor cells might adapt to this metabolic stress by activating antioxidant signaling pathways and acquire recurrence and dormancy (*Fox et al., 2020*; *Parida et al., 2022*). Although JIMT1 is resistant to trastuzumab, the extended exposure of trastuzumab was found to decrease SLC7A11 expression. SLC7A11 overexpression and altered cysteine and glutathione metabolism might function as compensating ways for trastuzumab adaptation. The dual inhibition of cysteine metabolism and HER2 also enhanced ferroptosis both in vivo and in vitro. In a similar manner, the inhibition of glutamine metabolism and antioxidant transcription factor NRF2 could also suppress the proliferation of residual breast cancer after HER2 inhibition and restore intracellular GSH metabolism (*Fox et al., 2020*). These results indicated that re-establishing redox homeostasis and modulating antioxidant metabolism might provide novel treatment options for trastuzumab primary-resistant HER2-positive breast cancer.

Besides amino acids metabolism, lipid metabolism, especially polyunsaturated fatty acids (PUFAs) and relevant phospholipid metabolism, which are also key drivers of ferroptosis (*Pope and Dixon, 2023*), were identified as dysregulated in both primary and acquired trastuzumab resistance of HER2-positive breast cancer (*Duan et al., 2024a*), demonstrating the possibility that targeting PUFA metabolism to induce ferroptosis might provide novel treatment options for overcoming trastuzumab resistance.

Not limited to cancer cells, altered cellular metabolism and oxidation-reduction balance can also affect activities of immune cells within TME (*De Martino et al., 2024*; *Leone and Powell, 2020*; *Chen et al., 2021*). For example, lipid peroxidation stress might impair NK functions and immune surveillance. Metabolites, such as L-kynurenine, might enhance ferroptosis in NK cells through aryl

hydrocarbon receptor and mediate immunosuppressive TME in gastric cancer (*Cui et al., 2023*). Activation of NRF2-related antioxidant pathways could restore NK cell functions and trigger enhanced anti-tumor responses (*Poznanski et al., 2021*). The deprivation of cysteine was reported to promote the metabolic reprogramming of TME and influence T cell immune responses (*Han et al., 2024*). We observed that the combination of cysteine starvation and trastuzumab can increase tumor-infiltrating NK cells but show limited effects on NK cell development and toxic functions, relevant mechanisms still need to be discussed. In addition, metabolites, cytokines, damage-associated molecular patterns (DAMPs) and other immunomodulatory signals released from cancer cells that undergo ferroptosis could function on immune cells nearby, stimulating or suppressing anti-tumor immune responses (*Dang et al., 2022*; *Zhou et al., 2024*).

The emergence of drug resistance in cancers mainly depends on two scenarios: Darwinian selection and Lamarckian induction (*Boumahdi and de Sauvage, 2020*; *Gatenby and Brown, 2020*). During the evolution of breast cancer, different subtypes of breast cancer might originate from different development status of mammary stem cells, bipotent progenitors, and differentiated cells (*Polyak, 2007*; *Chaffer and Weinberg, 2010*). HER2-positive breast cancer was reported to transform from luminal progenitor and mature luminal cells (*Lim et al., 2009*; *Prat and Perou, 2009*; *Tharmapalan et al., 2019*). The tumorigenesis of trastuzumab primary-resistant HER2-positive breast cancer might favor Darwinian selection, as a small population of pre-existing trastuzumab-tolerant cancer cells might be enriched upon treatment.

To investigate genetic and epigenetic issues stimulating the evolution of primary trastuzumab-resistant and sensitive breast cancer, we compared genomic variation, histone modification, and DNA methylation status among JIMT1, SKBR3, and MCF10A (normal mammary epithelial cell).

During carcinogenesis, missense mutations, splicing mutations, and in-frame deletions were the most frequent events and *MUC3A*, *MUC6*, *OR8U1*, and *ZNF717* were most frequent mutated genes (*Figure 6—figure supplement 3A–C*). Meanwhile, single nucleotide polymorphism (SNP) was the major variation type and the substitution between cytosine and thymine was the most common signature (*Figure 6—figure supplements 4A-C and 5A-C*).

Compared with MCF10A, JIMT1 instead of SKBR3 featured distinct single nucleotide variation (SNV) at the exon region of *SLC7A11* (gain at Chr4:138183261 and loss at Chr4: 138219340) (*Figure 6—figure supplement 6A–C*).

SNVs located in exons have been reported to modulate gene transcription and function. For example, SNVs in *BRCA1* exons encoding RING and BRCT domains might disrupt BRCA1 expression and function, therefore increasing breast cancer risks (*Findlay et al., 2018*). While noncoding SNVs, such as rs9383590 located at *ESR1* enhancer region, could interact with regulatory elements and activate oncogene transcription (*Bailey et al., 2016*). The accumulation of SNVs was also associated with driver gene mutations and might promote breast cancer evolution in a cumulative manner (*Nishimura et al., 2023*). Whether different SNVs in *SLC7A11* exon regions could contribute to SLC7A11 expression and breast cancer initiation still needs to be studied.

Epigenetic issues also demonstrated their impact on carcinogenesis. H3K4me3 instead of H3K27me3 peaks were shown more enrichment at *SLC7A11* promoter regions in MCF10A cells (*Figure 6—figure supplement 7A and B*), while 5-mC levels at CpG islands among *SLC7A11* promoter regions were also increased in MCF10A (*Figure 6—figure supplement 7C*), revealing the important proposal of histone modifications and DNA methylation on metabolic reprogramming during the evolution of HER2-positive breast cancer.

Although several studies have reported that SLC7A11 expression was modulated by NRF2, ATF4, and other transcription factors (*Koppula et al., 2021*; *Anandhan et al., 2020*; *Bai et al., 2021*), it remains unclear whether its expression is subject to a broader genome-wide or histone-level regulation (*Wang et al., 2023a*). The theory that epigenetic modifications contribute to specific transcription regulation, cancer cell development, and metabolic reprogramming, to modulate treatment response and drug resistance, has been widely accepted and proved (*Wang et al., 2023b*). Our study tried to explore the relationship between the altered cysteine metabolism signature and epigenetic modifications in trastuzumab-resistant HER2-positive breast cancer. Elucidating epigenetic mechanisms would help better understand trastuzumab resistance formation and provide novel therapeutic opportunities.

Several studies have described the regulatory roles of histone modifications on oncogene transcription (*Wang et al., 2023b*; *Sun et al., 2022*). MLL mutations were identified among various cancers and widespread shortening H3K4me3 peaks at tumor suppressor genes might contribute to tumor initiation (*Zhao et al., 2021*; *Chen et al., 2015*), and H3K27me3 could modulate chemotherapy tolerance in triple-negative breast cancer (TNBC) (*Marsolier et al., 2022*). Moreover, the combination of chemotherapy and H3K27me3 demethylation inhibitor could prevent the proliferation of persister cells (*Marsolier et al., 2022*). Targeting KDM6A could inhibit JIMT1 tumor growth via decreasing tRNA transcription (*Duan et al., 2024b*). As we confirmed, reducing H3K4me3 can decrease ASH2L, SLC7A11 expression, and cystine uptake, leading to enhanced lipid peroxidation and ferroptosis. Modulating histone modification can be a promising strategy in cancer treatment.

Compared with normal tissues, almost all cancers feature abnormal DNA hypomethylation and local hypermethylation at specific CpG islands (*Zhao et al., 2021*; *Nishiyama and Nakanishi, 2021*). Dysregulated DNA methylation contributes to breast cancer tumorigenesis and drug resistance formation (*Sher et al., 2022*; *Sukocheva et al., 2022*). For instance, trastuzumab-resistant HER2-positive breast cancer was found associated with downregulated TGFBI, CXCL2, and SLC38A1, which was silenced by DNA hypermethylation (*Palomeras et al., 2019*). Aberrant DNMT1, DNMT3A, and DNMT3B expression were correlated with endocrine therapy resistance and predicted worsened pathologic status and prognosis (*Jahangiri et al., 2019*). By utilizing CRISPR interference, we verified the effect of 5-mC at *SLC7A11* promoter on gene expression and downstream cysteine metabolism. What's more, since increased H3K4me3 might interact with DNMT3A-DNMT3L (*Du et al., 2015*), the crosstalk between histone modification and DNA methylation may have collaborative effects on cysteine metabolism in trastuzumab primary-resistant breast cancer.

In conclusion, our study revealed that trastuzumab primary-resistant HER2-positive breast cancer featured distinct cysteine metabolism, which was driven by altered H3K4me3 and DNA methylation. Targeting cysteine metabolism could provide novel treatment concepts for overcoming trastuzumab resistance.

## Acknowledgements

We would like to thank the Core Facility of the First Affiliated Hospital of Nanjing Medical University for its help in the detection of experimental samples. This project was supported by the National Natural Science Foundation of China (81972484 and 82272667 to Yongmei Yin; 82203488 to Ningjun Duan), High-level Innovation Team of Nanjing Medical University Program (JX102GSP201727), The Collaborative Innovation Center for Tumor Individualization Program (JX21817902/008) and Beijing Xisike Clinical Oncology Research Foundation (Y-2019AZZD-00680).

## Additional information

### Funding

| Funder | Grant reference number | Author |
| --- | --- | --- |
| National Natural Science Foundation of China | 81972484 | Yongmei Yin |
| National Natural Science Foundation of China | 82272667 | Yongmei Yin |
| National Natural Science Foundation of China | 82203488 | Ningjun Duan |
| High-level Innovation Team of Nanjing Medical University Program | JX102GSP201727 | Yongmei Yin |
| The Collaborative Innovation Center for Tumor Individualization Program | JX21817902/008 | Yongmei Yin |

| Funder | Grant reference number | Author |
|---|---|---|
| Beijing Xisike Clinical Oncology Research Foundation | Y-2019AZZD-00680 | Yongmei Yin |

The funders had no role in study design, data collection and interpretation, or the decision to submit the work for publication.

## Author contributions

Yijia Hua, Resources, Data curation, Software, Formal analysis, Validation, Investigation, Visualization, Methodology, Writing – original draft, Project administration, Writing – review and editing; Ningjun Duan, Conceptualization, Resources, Data curation, Software, Formal analysis, Supervision, Funding acquisition, Validation, Investigation, Visualization, Methodology, Writing – original draft, Project administration, Writing – review and editing; Chunxiao Sun, Fan Yang, Min Tian, Resources, Validation, Investigation, Methodology, Writing – review and editing; Yanting Sun, Resources, Data curation, Investigation, Methodology, Writing – review and editing; Shuhan Zhao, Data curation, Validation, Investigation, Methodology, Writing – review and editing; Jue Gong, Resources, Methodology, Writing – review and editing; Qian Liu, Resources, Investigation, Methodology, Project administration, Writing – review and editing; Xiang Huang, Resources, Supervision, Writing – review and editing; Yan Liang, Resources, Validation, Methodology, Project administration, Writing – review and editing; Ziyi Fu, Resources, Supervision, Investigation, Methodology, Writing – review and editing; Wei Li, Resources, Supervision, Validation, Writing – review and editing; Yongmei Yin, Conceptualization, Resources, Data curation, Software, Formal analysis, Supervision, Funding acquisition, Validation, Investigation, Visualization, Methodology, Project administration, Writing – review and editing

## Author ORCIDs

Yijia Hua ⓘ https://orcid.org/0009-0006-4323-8715
Ningjun Duan ⓘ https://orcid.org/0000-0001-6316-3796
Yongmei Yin ⓘ https://orcid.org/0000-0003-3335-369X

## Ethics

All blood samples were collected before the first trastuzumab treatment cycle. Samples collection was under the approval of the Ethics Committee and Institutional Review Board of Jiangsu Province Hospital. Each patient provided written informed consent for sample and data use.

All animal experiments were conducted according to the review and approval of Institutional Animal Care and Use Committee in Nanjing Medical University (IACUC-2204057).

Reviewer #1 (Public review): https://doi.org/10.7554/eLife.103953.3.sa1
Reviewer #2 (Public review): https://doi.org/10.7554/eLife.103953.3.sa2
Author response https://doi.org/10.7554/eLife.103953.3.sa3

# Additional files

## Supplementary files

Supplementary file 1. Oligonucleotide sequences and PCR primers sequences. Table 1. Oligonucleotide sequences of siRNAs. Table 2. Oligonucleotides sequences of small guide RNAs (sgRNAs). Table 3. PCR primers sequences for ChIP and MeDIP tests.

MDAR checklist

## Data availability

The metabolome, RNA sequencing, ChIP, WGBS and WGS data of cell lines have been deposited at figshare (https://doi.org/10.6084/m9.figshare.29120012.v1). The metabolome data of human samples generated by this study have been deposited at NGDC (National Genomics Data Center, https://ngdc.cncb.ac.cn/) with the BioProject PRJCA038837 (OMIX009855). Due to data privacy and ethical regulations, controlled-access data of human samples are available for non-commercial purposes upon request to the corresponding author. This article does not report original codes. Any further

information is available from the corresponding author upon request. Source data are provided with this paper.

The following datasets were generated:

| Author(s) | Year | Dataset title | Dataset URL | Database and Identifier |
|---|---|---|---|---|
| Yin Y | 2025 | Metabolic reprogramming in HER2 positive breast cancer | https://ngdc.cncb.ac.cn/bioproject/browse/PRJCA038837 | National Genomics Data Center, PRJCA038837 |
| Hua Y | 2025 | The metabolome, RNA sequencing, ChIP, WGBS and WGS data of JIMT1, SKBR3 and MCF10A | https://doi.org/10.6084/m9.figshare.29120012.v1 | figshare, 10.6084/m9.figshare.29120012.v1 |

The following previously published dataset was used:

| Author(s) | Year | Dataset title | Dataset URL | Database and Identifier |
|---|---|---|---|---|
| Wolf D, Yau C, van 't Veer L | 2021 | Neoadjuvant T-DM1/pertuzumab and paclitaxel/trastuzumab/pertuzumab for HER2-positive breast cancer in the adaptively randomized I-SPY2 trial | https://www.ncbi.nlm.nih.gov/geo/query/acc.cgi?acc=GSE181574 | NCBI Gene Expression Omnibus, GSE181574 |

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
