## [Editor Report · eLife Assessment]

This study provides **compelling** evidence that SLC7A11 may serve as a potential therapeutic target for trastuzumab-resistant HER2-positive breast cancer. While the findings are well-supported by robust data, the study could have been further strengthened by incorporating additional cell line experiments and providing more detailed clarification on patient sample selection. Nevertheless, this **valuable** work represents a significant contribution and will be of considerable interest to researchers in the field of breast cancer.

---

## [Referee Report · Reviewer #1 (Public review)]

Summary:

Hua et al show how targeting amino acid metabolism can overcome Trastuzumab resistance in HER2+ breast cancer.

Strengths:

The authors used metabolomics, transcriptomics and epigenomics approaches in vitro and in preclinical models to demonstrate how trastuzumab resistant cells utilize cysteine metabolism.

Weaknesses:

However, there are some key aspects that needs to be addressed.

Major:

(1) Patient Samples for Transcriptomic Analysis: It is unclear from the text whether tumor tissues or blood samples were used for the transcriptomic analysis. This distinction is crucial, as these two sample types would yield vastly different inferences. The authors should clarify the source of these samples.

(2) The study only tested one trastuzumab-resistant and one trastuzumab-sensitive cell lines. It is unclear whether these findings are applicable to other HER2-positive tumor cell lines, such as HCC1954. The authors should validate their results in additional cell lines to strengthen their conclusions.

(3) Relevance to Metastatic Disease: Trastuzumab resistance often arises in patients during disease recurrence, which is frequently associated with metastasis. However, the mouse experiments described in this paper were conducted only in the primary tumors. This article will have more impact if the authors could demonstrate that the combination of Erastin or cysteine starvation with trastuzumab can also improve outcomes in metastasis models.

Minor:

(1) The figures lack information about the specific statistical tests used. Including this information is essential to show the robustness of the results.

(2) Figure 3K Interpretation: The significance asterisks in Figure 3K do not specify the comparison being made. Are they relative to the DMSO control? This should be clarified.

Comments on revisions:

While the authors acknowledge the limitation of using only a single trastuzumab resistant/sensitive pair, simply stating that additional cell lines will be tested in future work is simply inadequate. The biological heterogeneity of HER2-positive breast cancer demands validation in at least one independent resistant model (e.g., HCC1954 or BT 474R) alongside its parental counterpart. Without demonstrating that SLC7A11 upregulation, cysteine dependency, and sensitivity to Erastin plus trastuzumab extend beyond the original cell line pair, the generalizability and translational relevance of the findings remain uncertain. The authors need to perform and report key functional results (cell viability, apoptosis, and SLC7A11 expression) in an additional resistant and sensitive HER2-positive cell line before this manuscript can be considered robust.

---

## [Referee Report · Reviewer #2 (Public review)]

In this manuscript, Hua et al. proposed SLC7A11, a protein facilitating cellular cystine uptake, as a potential target for the treatment of trastuzumab resistant HER2 positive breast cancer. If this claim holds true, the finding would be of significance and might be translated to clinical practice. Nevertheless, this reviewer finds that the conclusion was insufficiently supported by the data.

Notably, most of the data (Figures 2-6) were based on two cell lines - JIMT1 as a representative of trastuzumab resistant cell line, and SKBR3 as a representative of trastuzumab sensitive cell line. As such, these findings could be cell line specific while irrelevant to trastuzumab sensitivity at all. Furthermore, the authors' claim of ferroptosis induction is primarily based on lipid peroxidation assays (Figure 3). The rescuing effects of ferroptosis inhibitors on cell viability were missing. The xenograft experiments were also suspicious (Figure 4). Systemic cysteine starvation is known to cause adverse effects, including liver necrosis, and the compound (i.e., erastin) used by the authors is not suitable for in vivo experiments due to low solubility and low metabolic stability. Finally, the authors focus on epigenetic regulations (Figures 5 & 6) without first investigating well-established transcription factors, such as NRF2 and ATF4, which are known to regulate SLC7A11.

To sum up, this reviewer finds that the most valuable data in this manuscript is perhaps Figure 1, which provides unbiased information concerning the metabolic patterns in trastuzumab sensitive and primary resistant HER2 positive breast cancer patients.

Comments on revisions:

(1) Figure 3: The unit of concentration should be "μM". "μm" means micrometer.

(2) Figure S5: Ferroptosis inhibitors should be used in cell viability assays to exclude the off-target effect of RSL3 and erastin. Note that erastin also targets VDAC, while RSL3 may inhibit other selenoproteins at high concentrations. Cell viability assays are critical for demonstrating ferroptosis and should be included in the main figure rather than relegated to the supplemental materials.

(3) Figure 4B & 4C: the data of "H" group and "Erastin" group are inconsistent. In panel B, the tumor size in the "H" group appears smaller than in the "Erastin" group, while in panel C, the opposite trend is observed.

(4) The catalog numbers for the cystine/cysteine-deficient DMEM (from BIOTREE) and diet (from Xietong Bio) should be provided. This information is essential for readers to identify and verify the specific products used in the study.

---

## [Author Response]

The following is the authors’ response to the original reviews

**Public Reviews:**

**Reviewer #1 (Public review):**
Summary:Hua et al show how targeting amino acid metabolism can overcome Trastuzumab resistance in HER2+ breast cancer.Strengths:The authors used metabolomics, transcriptomics and epigenomics approaches in vitro and in preclinical models to demonstrate how trastuzumab-resistant cells utilize cysteine metabolism.

Thank you for your valuable comments. We would like to extend our appreciation for your efforts. Your constructive suggestion would help improve our research.

Weaknesses:However, there are some key aspects that needs to be addressed.Major:(1) Patient Samples for Transcriptomic Analysis: It is unclear from the text whether tumor tissues or blood samples were used for the transcriptomic analysis. This distinction is crucial, as these two sample types would yield vastly different inferences. The authors should clarify the source of these samples.

Thank you for your valuable comments. In the transcriptomic analysis, we included the data of HER2 positive breast cancer patients who received trastuzumab in I-SPY2 trial (GSE181574). Tumor tissues were used in this dataset. We highlighted the usage of “pre-treatment breast cancer tumors” in Line 309 and included the overview of transcriptomic data analysis in I-SPY2 trial in Figure S1F.

(2) The study only tested one trastuzumab-resistant and one trastuzumab-sensitive cell line. It is unclear whether these findings are applicable to other HER2-positive tumor cell lines, such as HCC1954. The authors should validate their results in additional cell lines to strengthen their conclusions.

Thank you for your valuable comments. We agree with your opinion, and the exploration of multiple cell lines would make our research findings more comprehensive. This is a limitation of our study, and we would continue to improve our design and methods in future experiments.

(3) Relevance to Metastatic Disease: Trastuzumab resistance often arises in patients during disease recurrence, which is frequently associated with metastasis. However, the mouse experiments described in this paper were conducted only in the primary tumors. This article would have more impact if the authors could demonstrate that the combination of Erastin or cysteine starvation with trastuzumab can also improve outcomes in metastasis models.

Thank you for your valuable comments. We agree with your suggestions. The exploration of metastatic disease would make our research more meaningful and help better address clinical key issues. In our future studies, we will continue to investigate the association between the invasive and metastatic capabilities of trastuzumab resistant HER2 positive breast cancer and cysteine metabolism.

Minor:(1) The figures lack information about the specific statistical tests used. Including this information is essential to show the robustness of the results.

Thank you for your valuable comments. We added statistical information in our figure legends, including Line 849-850, Line 865-867, Line 881-882, Line 898-900, Line 910-911 and Line 923-924.

(2) Figure 3K Interpretation: The significance asterisks in Figure 3K do not specify the comparison being made. Are they relative to the DMSO control? This should be clarified.

Thank you for your valuable comments. We have modified this figure to demonstrate it more clearly. In Figure 3K, the significance was determined by one-way ANOVA and the comparison presented was relative to the DMSO control. It was indicated that the combination of erastin or cysteine starvation and trastuzumab could increase lipid peroxidation, although trastuzumab monotherapy did not induce ferroptosis.

Additionally, the combination of erastin and trastuzumab could result in more lipid peroxidation than erastin alone. Similar results were also found in the combination of cysteine starvation and trastuzumab. These results showed that targeting cysteine metabolism plus trastuzumab could have synergic effects to induce ferroptosis in trastuzumab resistant HER2 positive breast cancer.

**Reviewer #2 (Public review):**
In this manuscript, Hua et al. proposed SLC7A11, a protein facilitating cellular cystine uptake, as a potential target for the treatment of trastuzumab-resistant HER2-positive breast cancer. If this claim holds true, the finding would be of significance and might be translated to clinical practice. Nevertheless, this reviewer finds that the conclusion was poorly supported by the data.Notably, most of the data (Figures 2-6) were based on two cell lines - JIMT1 as a representative of trastuzumab-resistant cell line, and SKBR3 as a representative of trastuzumab sensitive cell line. As such, these findings could be cell-line specific while irrelevant to trastuzumab sensitivity at all. Furthermore, the authors claimed ferroptosis simply based on lipid peroxidation (Figure 3). Cell viability was not determined, and the rescuing effects of ferroptosis inhibitors were missing. The xenograft experiments were also suspicious (Figure 4). The description of how cysteine starvation was performed on xenograft tumors was lacking, and the compound (i.e., erastin) used by the authors is not suitable for in vivo experiments due to low solubility and low metabolic stability. Finally, it is confusing why the authors focused on epigenetic regulations (Figures 5 & 6), without measuring major transcription factors (e.g., NRF2, ATF4) which are known to regulate SLC7A11.To sum up, this reviewer finds that the most valuable data in this manuscript is perhaps Figure 1, which provides unbiased information concerning the metabolic patterns in trastuzumab-sensitive and primary resistant HER2-positive breast cancer patients.

Thank you for your valuable comments. We agree with your suggestions. Your feedback would help enhance the quality of our research.

(1) Our research was mainly conducted in JIMT1 (trastuzumab resistant) and SKBR3 (trastuzumab sensitive), and this is a limitation of our study. The experimental validation using different cell lines will make our research findings more persuasive. In our future research, we will continuously optimize experimental design and methods to make our findings more comprehensive.

(2) The detection of ferroptosis in our research was mainly performed by evaluating the lipid peroxidation. Experiments measuring cell viability and rescuing effects would help provide more evidence.

We utilized CCK8 tests to compare cell viabilities of JIMT1 and SKBR3 in different erastin and RSL3 concentrations, as well as different exposure time of cysteine starvation. It was shown that JIMT1 was more sensitive to erastin and RSL3, but tolerant to cysteine starvation, which was consistent with the previous lipid peroxidation tests. This data was included in Figure S5C-E. We added the description in Line 375-379.

In addition, we also performed experiments to explore the rescuing effects of ferroptosis inhibitor Fer-1. It was indicated that Fer-1 could suppress the lipid peroxidation resulted from erastin, RSL3 and cysteine starvation in both JIMT1 and SKBR3. This provided more evidence that cysteine metabolism played a vital role in modulating HER2 positive breast cancer ferroptosis. This data was included in Figure S5G and S5H. We added the description to Line 387-391.

(3) In xenograft experiments, the cysteine starvation was performed by feeding cystine/cysteine-deficient diet (Xietong Bio). We added details of this diet on Line 236-237 in Methods.

We agree with your opinion on the role of erastin in experiments in vivo. We have tried to optimize drug dissolution and other conditions by referring to previous relevant literature. We would continue to improve our experimental design and methods.

(4) Epigenetic modifications have been recognized as crucial factors in drug resistance formation. An increasing number of studies have emphasized the importance of epigenetic changes in regulating the abnormal expression of oncogenes and tumor suppressor genes related to drug resistance. Currently, the role of epigenetic changes in the development of trastuzumab resistance in HER2 positive breast cancer is still in exploration. We tried to investigate the dysregulation of histone modifications and DNA methylation in trastuzumab resistant HER2 positive breast cancer. Our findings indicated that targeting H3K4me3 and DNA methylation could decrease SLC7A11 expression and induce ferroptosis. This would provide more evidence in exploring trastuzumab resistance mechanisms. We have provided a detailed discussion on Line 598-607.

We would like to extend our appreciation for your constructive suggestions and continue to improve our research in future experiments.

**Recommendations for the authors:**

**Reviewer #2 (Recommendations for the authors):**
(1) Line 334: it would be helpful to clarify that JIMT1 cells are trastuzumab-resistant while SKBR3 cells are trastuzumab sensitive, especially for those not familiar with breast cancer cell lines.

Thank you for your valuable recommendations. We added the description of trastuzumab sensitive SKBR3 and trastuzumab resistant JIMT1 on Line 334-335.

(2) Figure 3: the concentrations of erastin and RSL3 should be indicated.

Thank you for your valuable recommendations. In Figure 3, the concentration of erastin was 10μm and RSL3 was 1μm. We added these details in the figure legends on Line 872-873.

(3) Figure 3: lipid peroxidation does not necessarily mean ferroptosis. Cell viability data and rescuing effects of ferroptosis inhibitors should be shown.

Thank you for your valuable recommendations. As we mentioned above, we utilized CCK8 tests to compare cell viabilities of JIMT1 and SKBR3 in different erastin and RSL3 concentrations, as well as different exposure time of cysteine starvation. It was consistent with lipid peroxidation tests that JIMT1 was more sensitive to erastin and RSL3, but tolerant to cysteine starvation. This data was included in Figure S5C-E. We added the description in Line 375-379.

As described above, we also performed experiments to explore the rescuing effects of ferroptosis inhibitor Fer-1. It was indicated that Fer-1 could suppress the lipid peroxidation resulted from erastin, RSL3 and cysteine starvation in both JIMT1 and SKBR3. This provided more evidence that cysteine metabolism played a vital role in modulating HER2 positive breast cancer ferroptosis. This data was included in Figure S5G and S5H. We added the description to Line 387-391.

(4) Figure 3H: how cysteine starvation was performed should be clarified in the Methods section.

Thank you for your valuable recommendations. We performed cell culture with cysteine starvation by utilizing cystine/cysteine-deficient DMEM (BIOTREE) and 1% penicillin streptomycin at 37℃ with 5% CO2. We added details of this diet on Line 141-143 in Methods.

(5) Figure 4: the meaning of "H" should be clarified.

Thank you for your valuable recommendations. H was indicated as trastuzumab. We clarified the meaning of “H” in the figure legends on Line 898.

(6) Figure 4B & 4C: the data of "H" group and "Erastin" group are inconsistent.

Thank you for your valuable recommendations. In the vivo experiments, the tumor volume changes were analyzed using a paired approach, comparing the tumor size of each individual mouse before and after treatment. We noticed the confusion caused and added more details about our vivo experiments on Line 240 in Methods and Line 892-893 in figure legends.

(7) Figure 4: how cysteine starvation was performed should be clarified in the Methods section.

Thank you for your valuable recommendations. We performed cysteine starvation by utilizing cystine/cysteine-deficient diet (Xietong Bio). We added details of this diet on Line 236-237 in Methods.

We have also corrected some grammatical errors in the manuscript and We would like to extend our great appreciation to all editors and reviewers for their invaluable contributions.